# Formation of ammonia–helium compounds at high pressure

Jingming Shi [1], Wenwen Cui [1], Jian Hao[1,2], Meiling Xu[1], Xianlong Wang [3✉] & Yinwei Li [1✉]

Uranus and Neptune are generally assumed to have helium only in their gaseous atmospheres. Here, we report the possibility of helium being fixed in the upper mantles of these planets in the form of $NH_3$–He compounds. Structure predictions reveal two energetically stable $NH_3$–He compounds with stoichiometries $(NH_3)_2He$ and $NH_3He$ at high pressures. At low temperatures, $(NH_3)_2He$ is ionic with $NH_3$ molecules partially dissociating into $(NH_2)^-$ and $(NH_4)^+$ ions. Simulations show that $(NH_3)_2He$ transforms into intermediate phase at 100 GPa and 1000 K with H atoms slightly vibrate around N atoms, and then to a superionic phase at ~2000 K with H and He exhibiting liquid behavior within the fixed N sublattice. Finally, $(NH_3)_2He$ becomes a fluid phase at temperatures of 3000 K. The stability of $(NH_3)_2He$ at high pressure and temperature could contribute to update models of the interiors of Uranus and Neptune.

[1] Laboratory of Quantum Materials Design and Application, School of Physics and Electronic Engineering, Jiangsu Normal University, Xuzhou 221116, China. [2] Jiangsu Key Laboratory of Advanced Laser Materials and Devices, Jiangsu Normal University, Xuzhou 221116, China. [3] Key Laboratory of Materials Physics, Institute of Solid State Physics, Chinese Academy of Sciences, Hefei 230031, China. ✉email: xlwang@theory.issp.ac.cn; yinwei_li@jsnu.edu.cn

Knowledge of the interior compositions of planets is crucial to understanding the processes of their formation and evolution. Various methods have been used to investigate the Earth's interior, while studies of the composition and structure of the solar system's ice giants, Uranus and Neptune, are limited by using only global observable properties such as gravitational and magnetic moments[1]. Uranus and Neptune are generally assumed to have a three-layer structure: a rocky core, an ice mantle (contains an upper mantle and a lower mantle), and a gas atmosphere[1–6]. Although many studies have focused on the interiors of Uranus and Neptune, their internal compositions remain to be fully understood[7–12]. A widely accepted model for each of these planets is that the upper mantle comprises a mixture of ionized $H_2O$, $NH_3$, and $CH_4$[5,8,11], whereas the lower mantle consists of metallic $H_2O$, $NH_3$[5,6] Much effort has been devoted to determine the ratio of the components in the interior of the ice planets[8,10,11], however, no consensus were reached.

To understand realistic compositions of Uranus and Neptune, researchers have focused on high-pressure and high-temperature phases of $NH_3$ and $H_2O$, and mixtures of the two[13–15]. Cavazzoni et al[16]. performed molecular dynamic simulations to estimate the phase diagram of water and ammonia at pressures and temperatures in the range of 30– 300 GPa and 300–7000 K. They found that water and ammonia exhibited a superionic phase (at about 100 GPa, 1500 K) between the ionic solid phase and ionic fluid phase. However, for the component $CH_4$, the results of equation of state have shown that it would dissociate into hydrocarbons at the extreme conditions[17]. A computational search undertaken by Pickard and Needs in 2008 found that ammonia transformed into an ionic phase consisting of $(NH_2)^-$ and $(NH_4)^+$ ions at pressures above 90 GPa[13]; the transformation was subsequently confirmed by experiment[14,15]. By combining empirical and theoretical results, Ninet et al[18]. found that a novel superionic conductive phase of ammonia becomes stable at about 70 GPa and 8500 K. Using Raman spectroscopy and synchrotron X-ray diffraction, Laniel et al[19]. found two unusual ionic N–H stoichiometries, $(N_2)_6(H_2)_7$ and $N_2(H_2)_2$, which are stable at about 50–GPa. While for water, it has a rich phase diagram, with at least 17 solid phases identified experimentally[20–22], and seven other high-pressure phases predicted by theoretical studies[23–26]. Ninet et al[18]. and Millot et al[27]. proposed that superionic water ice can exist in the mantles of the ice giants as a result of shock compression. Recently, Huang et al[28]. found that $H_2O$ can react with $H_2$ and form a novel superionic compound of $H_3O$ under high pressure and high temperature.

For a 2:1 mixture of $NH_3$ and $H_2O$, Robinson et al[29]. predicted a novel ionic compound, $O_2^-(NH_4^+)_2$, to form at pressures above 65 GPa. Recent theoretical and experimental studies have shown that $NH_3H_2O$ decomposes into ammonia and water at 120 GPa[30,31]. Bethkenhagen et al[32]. used an evolutionary random structure search code to propose a superionic phase of $NH_3H_2O$ at 800 GPa and high temperature (1000–6000 K). An unusual layered ionic phase of $NH_3(H_2O)_2$ was predicted for a 1:2 mixture of $NH_3$ and $H_2O$; it was then modeled to transform into a superionic phase at high pressure and high temperature (41 GPa and 600 K)[33]. These findings contribute to our understanding of the interiors of the giant ice planets.

The above results have led to the assumption that the elements (i.e., C, H, and N) in the ice giants' gaseous atmospheres except He appear in their solid mantles. Helium is generally considered likely to remain only in the atmosphere and not form solid compounds in the mantle, because it is the most chemically inert element due to its stable closed-shell electronic configuration. In fact, Nettelmann et al[11]. have proposed a three-layer structure model, in which the considering of small amount of He/H in the outer core of the planets reproducing well the gravitational moments of the ice giants. Recent studies have indicated that high pressure can induce He to form compounds such as $HeN_4$[34], $Na_2He$[35], $FeHe$[36], $MgOHe$[37], $H_2OHe$[38,39], and $FeO_2He$[40]. Specially, the compound of $H_2OHe_2$ exhibited a superionic property under high pressure and high temperature and then transformed into fluid[39]. These results inspired us to explore whether some of the abundant elemental He from the planets' atmospheres could be trapped inside the mantles of Uranus and Neptune. Therefore, we carried out calculations to search for stable compounds in $NH_3$–He systems at high pressure and high temperature. Our results show that He can react with $NH_3$ to form $(NH_3)_2He$ under extreme conditions, to a certain extent corresponding to the upper mantles of Uranus and Neptune, thereby providing information essential to the understanding of the interior models of these planets.

## Results

**Stable $NH_3$–He compounds at high pressure**. The formation enthalpies of the energetically most-stable structures of $(NH_3)_xHe_y$ ($x = 1 \sim 3$, $y = 1 \sim 3$) as compared to mixtures of $NH_3$ and He at selected pressures are summarized in Fig. 1. The phases lying on the convex hull are thermodynamically stable against decomposition into other compositions. The figure also shows the effects of zero-point vibrational energy (ZPE). The positive formation enthalpies show that, as expected, no thermodynamically stable compositions were found at ambient pressure. However, static-lattice enthalpy calculations revealed three stable compositions at high pressures: $(NH_3)_2He$ at 10 and 300 GPa, $NH_3He$ at 50, 100, and 150 GPa, and $NH_3He_2$ at 300 GPa (Supplementary Fig. 1). The inclusion of ZPE alters significantly the stability of $(NH_3)_2He$ and $NH_3He_2$, i.e., $(NH_3)_2He$ becomes energetically stable also at 150 and 300 GPa, while $NH_3He_2$ turns to be unstable at all pressures.

Figs. 2 and 3, respectively present detailed stable pressure ranges for the three obtained compositions and their corresponding crystal structures. Optimized lattice parameters for all the structures at selected pressures are listed in Supplementary Table 1. The results indicate that $(NH_3)_2He$ with space group $I4$, labeled here as $I4$-$(NH_3)_2He$, becomes energetically stable with respect to $NH_3$ and He at pressures as low as 9 GPa (Fig. 2). Tetragonal $I4$-$(NH_3)_2He$ consists of isolated $NH_3$ molecules and He atoms. Figure 3a depicts the $NH_3$ layers of this structure in the $a$–$b$ planes, with He atoms located in the pockets formed by neighboring $NH_3$ molecules. Interestingly, the $I4$ structure displays unique channels formed by $NH_3$ molecules that are arranged parallel to the $c$-axis, and linear He chains localize within the interstices formed by four neighboring channels (Fig. 3b). To our knowledge, this is the first report of such a channel-bearing $NH_3$ structure.

$(NH_3)_2He$ remains energetically stable up to 40 GPa, above which it decomposes into a mixture of $NH_3$ and He. However, $(NH_3)_2He$ re-emerges as energetically stable phase at 110 GPa with the formation of an orthorhombic $Fmm2$ structure. A similar combination-decomposition-recombination pattern has previously been reported for $CaLi_2$[44]. Partial dissociation of $NH_3$ molecules into $(NH_2)^-$ and $(NH_4)^+$ ions is found to accompany the formation of $Fmm2$-$(NH_3)_2He$. Bader analysis demonstrates the strongly ionic nature of the species, with Bader charges of $-0.57$ $e^-$ and 0.53 $e^-$ for $(NH_2)^-$ and $(NH_4)^+$, respectively, similar to those observed in the ionic phase of pure $NH_3$[13]. The $Fmm2$ phase is also layered, consisting of layers formed by $NH_3$, $(NH_2)^-$, and $(NH_4)^+$ units in the $a$–$c$ planes. The spacing between neighboring layers is 2.07 Å. Viewing the structure along the $a$-axis reveals unique channels formed by $NH_3$, $(NH_2)^-$, and $(NH_4)^+$ units, with He atoms located in the interstices.

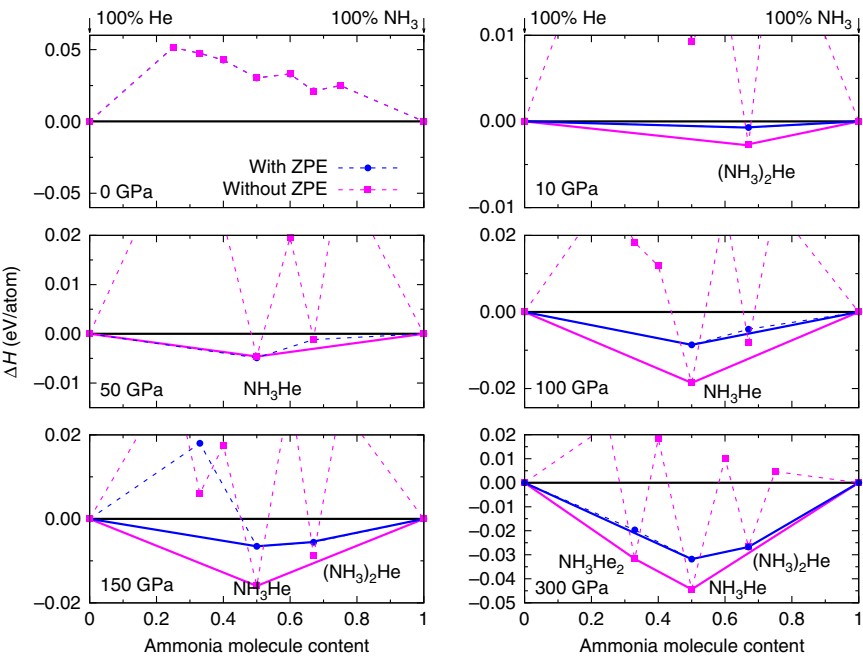

**Fig. 1 Stability of the NH₃-He system.** Calculated enthalpies of formation ($\Delta H$) of various NH₃–He compounds with respect to decomposition into solid NH₃ and He at selected pressures. Convex hulls are shown as solid lines with (blue) and without (magenta) the inclusion of zero-point vibrational energies. Calculations adopted the $P2_1/3$ structure[41] at 10 GPa, the $P2_12_12_1$ phase at 50 GPa, and the $Pma2$ structure at 100, 150, and 300 GPa for NH₃[13] and $hcp$-He[42,43] at all pressures.

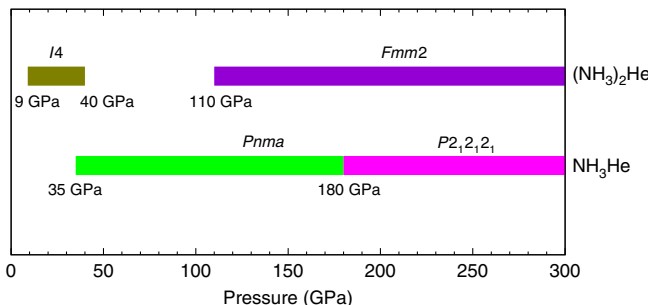

**Fig. 2 Pressure-composition phase diagram.** Phase diagram showing stable NH₃−He compounds with respect to pressure at 0 K. Colored regions represent the stable pressure ranges of each composition with respect to a mixture of NH₃ and He. Yellow-green, violet, green and pink represent $I4$-(NH₃)₂He, $Fmm2$-(NH₃)₂He, $Pnma$–NH₃He and $P2_12_12_1$–NH₃He phase, respectively.

NH₃He composition becomes energetically stable at 35 GPa, as shown in Fig. 2, adopting an orthorhombic $Pmma$ structure. The $Pmma$ phase is the most stable configuration over a large pressure range up to 180 GPa for NH₃He, above which it transforms into the $P2_12_12_1$ structure. The $P2_12_12_1$–NH₃He phase will remain stable up to 300 GPa, which is the maximum pressure considered in this study. Both the $Pmma$ and the $P2_12_12_1$ structures of NH₃He are composed of isolated NH₃ molecules and He atoms, and there is no evidence of dissociation of NH₃ in the whole pressure range studied here. The calculated phonon dispersions confirm the dynamical stability of all these structures in their energetically stable pressure ranges (Supplementary Fig. 3).

**Electronic properties.** To examine the interactions among N, H, and He atoms in the two compounds, we calculate electronic properties including the electronic localization function (ELF) and Bader charges. The ELF is a quantum chemistry tool to visualize covalent bonds; values close to 1 corresponding to strong covalent bonding. The ELF results rule out covalent bonds between N–H units (NH₃, (NH₂)⁻, (NH₄)⁺) and He atoms, given the absence of any ELF local maxima between them (Supplementary Fig. 5). Interestingly, Bader analysis indicates a slight charge transfer from N–H units to He atoms. Table 1 lists the Bader charge of one He atom in $I4$-(NH₃)₂He as ~ -0.02 $e^-$ at 10 GPa, which increases to -0.03 $e^-$ when the $Fmm2$ structure is adopted at 120 GPa. Similar to that in (NH₃)₂He, each He atom in NH₃He and NH₃He₂ gains nearly 0.03 $e^-$ from the NH₃ molecules. The Bader charge of a He atom in the three NH₃–He compounds is similar to the charges predicted for H₂O–He, MgF₂He, MgOHe, and FeO₂He (between −0.02 $e^-$ and -0.07 $e^-$)[36–38,40]. The current results indicate the three compounds have an ionic nature and that He atoms could serve as a Coulomb shield in stabilizing them at high pressure. Electronic band structures show that all three compounds are insulators (Supplementary Fig. 4). At the PBE-GGA level, the band gap of (NH₃)₂He is calculated to be 6.0 eV at 10 GPa, which increases to 7.5 eV at 180 GPa. For NH₃He, the band gaps is calculated to be 7.2 eV at 35 GPa.

**Superionic phases of (NH₃)₂He.** The stable pressure and temperature regions of $Fmm2$-(NH₃)₂He cover the geotherms in the upper mantle of Neptune and Uranus. We, therefore, performed ab initio molecular dynamics simulations at the pressure of 100 GPa, 200 GPa, and 300 GPa, respectively, to examine the formation of $Fmm2$–NH₃)₂He inside Neptune and Uranus. The calculated mean squared displacement (MSD) of the atomic positions and the behaviors of three different atoms of $Fmm2$-(NH₃)₂He are shown in the Fig. 4. At $P = 100$ GPa and $T = 200$ K, $Fmm2$-(NH₃)₂He is a solid phase with all atoms vibrating around their lattice positions and with diffusion coefficients ($D^H = D^{He} = D^N = 0$). When the temperature increasing to 1000 K, the H atoms seems diffusive with

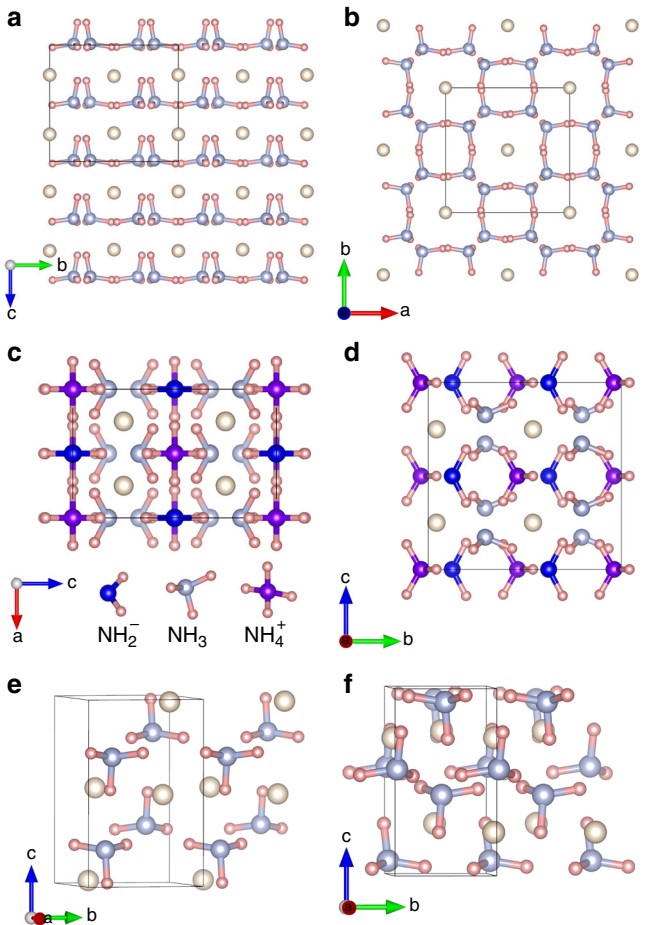

**Fig. 3 Structural configurations. a**, **b** The *I4* structure of $(NH_3)_2$He at 10 GPa along the *a*-axis and *c*-axis, respectively. **c**, **d** The *Fmm2* structure of $(NH_3)_2$He at 180 GPa in the direction of *b*-axis and *a*-axis, respectively. **e** The *Pnma* structure of $NH_3$He at 30 GPa, and **f** the $P2_12_12_1$ structure of $NH_3$He at 180 GPa. The light blue, pink, and cream-colored atoms represent N, H, and He, respectively. Blue, light blue, and purple spheres in **c**, **d** are used to distinguish the N atoms in $(NH_2)^-$, $NH_3$, and $(NH_4)^+$, respectively.

**Table 1 Bader charges.**

| Phase | Pressure (GPa) | Atom/ Unit | Charge ($e^-$) |
|---|---|---|---|
| *I4*-$(NH_3)_2$He | 10 | $NH_3$ | 0.01 |
| | | He | −0.02 |
| *Fmm2*-$(NH_3)_2$He | 120 | $NH_2$ | −0.57 |
| | | $NH_3$ | 0.05 |
| | | $NH_4$ | 0.53 |
| | | He | −0.03 |
| *Pnma*-$NH_3$He | 40 | $NH_3$ | 0.03 |
| | | He | −0.03 |
| $P2_12_12_1$-$NH_3$He | 180 | $NH_3$ | 0.03 |
| | | He | −0.03 |

Bader charges of the *I4* phase of $(NH_3)_2$He at 10 GPa, the *Fmm2* phase of $(NH_3)_2$He at 120 GPa, the *Pnma* phase of $NH_3$He at 40 GPa, and the $P2_12_12_1$ phase of $NH_3$He at 180 GPa. A negative (positive) sign indicates an electron gain (loss) for the particular atom or molecule.

$D^H = 1.4 \times 10^{-6}$ cm$^2$ s$^{-1}$. However, from the atomic trajectories shown in Fig. 4b, one can find that H atoms in $NH_3$ become diffuse while H atoms in $(NH_2)^-$ and $(NH_4)^+$ keep vibrating around their lattice positions. This means that the H

atoms in $NH_3$ units become considerable vibrate with a fixed N position at this condition. With the temperature further increased to 2000 K, *Fmm2*-$(NH_3)_2$He transforms into a real superionic phase with fully diffusive H atoms ($D^H = 2.0 \times 10^{-4}$ cm$^2$ s$^{-1}$) within the fixed N and He framework. With the temperature increased to 3000 K, all atoms including N, He, and H are diffusive with high diffusion coefficients ($D^N = 4.4 \times 10^{-5}$ cm$^2$ s$^{-1}$, $D^{He} = 2.1 \times 10^{-5}$ cm$^2$ s$^{-1}$ and $D^H = 4.5 \times 10^{-4}$ cm$^2$ s$^{-1}$). This result reveals that at this conditions the superionic *Fmm2*-$(NH_3)_2$He phase transformed into a fluid phase. Here, we found the diffusion of H atoms occurs prior to that of He atom, which is opposite to that found for He$_2$(H$_2$O)[39], where He atoms diffuse firstly. Generally, lighter atoms are easier to diffuse. The abnormal diffusive behavior in He$_2$(H$_2$O) was explained by that the H atoms has higher diffusion barrier than He atoms because of the strong covalent H-O bonds[39]. In fact, He atoms in He$_2$(H$_2$O) share large space that allows the free diffusion, as shown in Supplementary Fig. 6. As compared to He$_2$(H$_2$O), although form weak interaction with N–H units, He atoms are trapped in cages formed by $NH_3$, $(NH_2)^-$ and $(NH_4)^+$ units, this makes helium atoms are more difficult to diffuse.

While for $P = 200$ GPa and $T = 300$ K, *Fmm2*-$(NH_3)_2$He keeps its solid property. With the temperature increasing to 1000 K and up to 4000 K, *Fmm2*-$(NH_3)_2$He becomes to a superionic phase and then turns in to a fluid when the temperature is above 4200 K, as shown in Supplementary Fig. 7. Under pressure of 300 GPa, the trend is similar to that under 200 GPa, but the critical point of the superionic phase to fluid is at the temperature of 4600 K, as shown in Supplementary Fig. 8. Figure 5 presents the pressure–temperature (P–T) phase diagram for the mixture of $NH_3$ and He, showing the $(NH_3)_2$He and $NH_3$He phases. Temperature has a significant effect on the system: *I4*-$(NH_3)_2$He and $NH_3$He decompose at high temperature (Fig. 5 and Supplementary Fig. 2). Their maximum temperatures of stability vary, being >700 K for *I4*-$(NH_3)_2$He (which decomposes fully to $NH_3$ and He), >1000 K for $NH_3$He (for full decomposition to $NH_3$ and He at $P < 100$ GPa and decomposition into *Fmm2*-$(NH_3)_2$He and He at $P > 100$ GPa). In contrast, *Fmm2*-$(NH_3)_2$He has a large stability field and thermodynamically stable in pressure range of 80–300 GPa and at any temperature in the tested range (0–5000 K). Figure 5 also presents estimated geotherms for the interiors of Uranus and Neptune. We also pointed the phase states of *Fmm2*-$(NH_3)_2$He in the Fig. 5. Our calculation show that the *Fmm2*-$(NH_3)_2$He phase presents superionic and fluid properties at the condition which is close to the geotherms in the upper mantle of Neptune and Uranus. This suggests that He could be trapped as superionic $(NH_3)_2$He inside the upper mantles of these planets with the mixture of superionic and fluid forms during their formation.

Previous studies have assumed the presence of $NH_3$, $CH_4$, $H_2O$, and $H_2$ inside the giant ice planets. Our predicted stability of superionic $(NH_3)_2$He as well as the recent reported superionic $H_2O$He$_2$[39] under the P–T conditions corresponding to the ice giants' upper mantles indicate that helium could be remained inside the planets during their formation. Coincidently, the stability of $NH_3$–He and $H_2O$–He compounds provide an evidence to support the new three-layer model suggested by Nettelmann[11], in which helium was considered as a small component in outer core of the planets. Therefore, the current results are essential to the understanding of the interior models of these planets. Moreover, $CH_4$ and $H_2$ are another two main components in upper mantle of these planets, therefore, there is a high possibility that helium could react with $CH_4$ or $H_2$ at high pressures to form new compounds, which deserves further investigation.

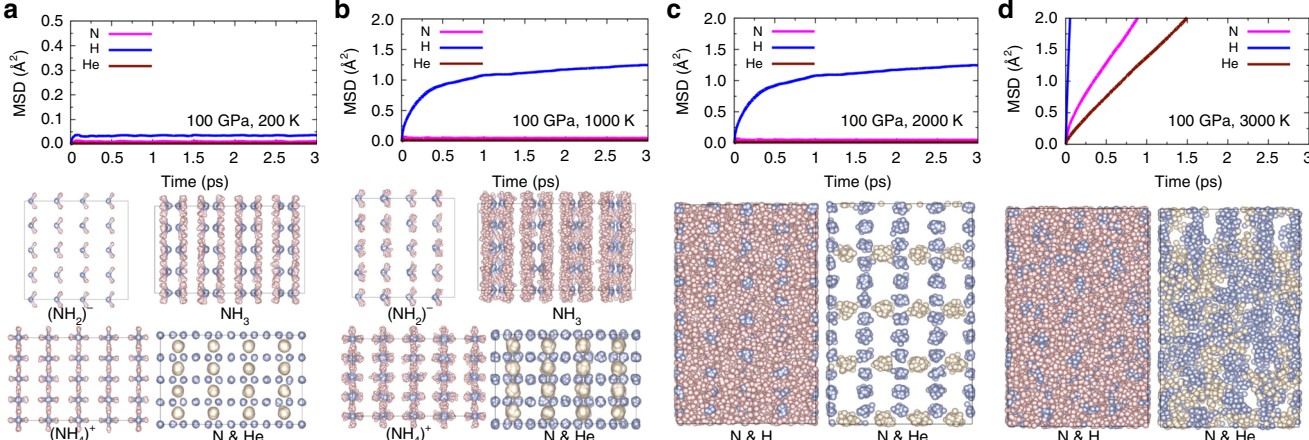

**Fig. 4 Dynamical behavior and atomic trajectories.** The calculated mean squared displacement (MSD) of the atomic positions of $Fmm2$-$(NH_3)_2He$ phase at the pressure of 100 GPa and the temperatures of **a** 200 K, **b** 1000 K, **c** 2000 K, and **d** 3000 K. The behaviors of different units (($NH_2$)$^-$, $NH_3$, and ($NH_4$)$^+$) or three different atoms (N, H, and He) in the molecular dynamics simulations are shown in different conditions. The light blue, pink, and cream-colored atoms represent N, H, and He, respectively.

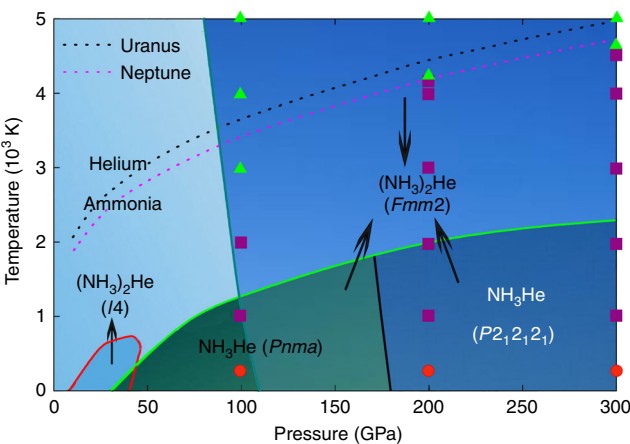

**Fig. 5 Pressure–temperature phase diagram.** Solid lines represent the stability field of each composition. Dashed lines indicate planetary geotherms estimated for the interiors of Uranus (black) and Neptune (magenta)[45]. Molecular dynamic calculations are performed under different extreme conditions and the results are marked in the figure: The red circle, purple square, and green triangle represent solid phase, superionic phase and fluid, respectively. Black arrows indicate the stable pressure–temperature regions associated with the arrowhead pointing phases.

In our submission process, we were aware of the work by Liu et al.[46] predicting plastic and superionic helium-ammonia compounds at extreme condition. They predicted three stable stoichiometries and eight new stable phases of He–$NH_3$ compounds under pressures up to 500 GPa and found that the predicted He–$NH_3$ compounds exhibit superionic behavior at high pressure and high temperature. These similar results further provide knowledge for our understanding of the composition of the planet's interior.

In summary, a combination of first-principles calculations and crystal structure predictions was carried out to search for stable compounds in the $NH_3$–He systems under high-$P$–$T$ conditions. Calculations at 0 K revealed two compounds (($NH_3$)$_2$He and $NH_3$He) that are energetically stable relative to the equivalent mixture of solid $NH_3$ and He at high pressures. Specially,

$(NH_3)_2He$ remains energetically stable under the extreme conditions corresponding to the upper mantles of Uranus and Neptune. The current results provide evidence that He could be trapped inside these planets as $NH_3$–He compounds with the mixture of superionic and fluid properties, in contrast to the current view that He occurs only in their atmospheres. Molecule dynamic simulations results show that the $Fmm2$-$(NH_3)_2He$ phase will transform into a superionic solid and then to a fluid with the increasing temperature.

## Methods

**Structural predictions**. Structure predictions for $NH_3$–He compounds were performed using a particle-swarm optimization algorithm implemented in CALYPSO code[47,48]. This method is unbiased, not using any known structural information, and has successfully been used to predict various systems under high pressure[49–57]. We performed structural searches on $(NH_3)_x He_y$ ($x, y = 1, 2, 3$) at 0–300 GPa with maximum simulation cells up to four formula units. Each generation of structures was evolved by selecting the 60% lowest-enthalpy structures in the last step and randomly producing the remaining 40%. The structure searches were considered converged when ~1000 successive structures were generated without finding a new lowest-enthalpy structure.

**Ab initio calculations**. Density functional theory calculations were performed using VASP code[58] combined with the generalized gradient approximation (GGA)[59] for the exchange-correlation potential in the form of the Perdew–Burke--Ernzerhof[60] (PBE) functional. The electronic wave functions were expanded in a plane wave basis set with a cutoff energy of 1000 eV. The electronic interaction was described by means of projector augmented wave[61] pseudopotentials with valence electrons of $1s^1$, $2s^2 2p^3$ and $1s^2$ for H, N, and He atoms, respectively. Monkhorst-Pack k-point[62] meshes with a grid density of 0.03 Å$^{-1}$ were chosen to achieve a total energy convergence of better than 1 meV per atom. The phonon dispersion curves were computed by direct supercell calculation[63], as implemented in the PHONOPY program[64].

**Molecular dynamics**. The molecular dynamics simulations were also carried out to explore the superionic property of $(NH_3)_2He$ compound at high pressures and high temperatures. The simulation supercells contain 32 $NH_3$ molecules and 16 helium atoms and the Brillouin zone was sampled by Γ point. Each simulation consists of 10,000 time steps with a time step of 0.5 fs.

## Data availability

The authors declare that the main data supporting the findings of this study are contained within the paper and its associated Supplementary Information. All other relevant data are available from the corresponding author upon reasonable request.

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

## Acknowledgements

The authors acknowledge funding from the NSFC under grant No. 11804129, No. 11722433, No. 11804128, No. 11904142 and No. 11674329. Y.L. acknowledges funding from the Six Talent Peaks Project of Jiangsu Province. X.W. acknowledges project No. TZ2016001 of Science Challenge. All the calculations were performed at the High Performance Computing Center of the School of Physics and Electronic Engineering of Jiangsu Normal University. Crystal structures were visualized with VESTA[65].

## Author contributions

J.S and Y.L. designed the project. J.S. and W.C performed the calculations. J.S., W.C., X.W., M.X., J.H., and Y.L. analyzed the data. J.S., W.C., X.W., and Y.L. wrote the paper.

## Competing interests

The authors declare no competing interests.
