## [Peer Review File · Nature Communications]

REVIEWERS' COMMENTS:

Reviewer #1 (Remarks to the Author):

The major claim of this paper is that there exists solid solutions of ammonia and helium at the pressure and temperature relevant to Ice Giants such as Neptune and Uranus and that it can support a super-ionic phase. The study is well constructed although somewhat derivative in that they basically follow the outline given in Bethkenhagen et al. (2015), i.e. a combination of structure search, quasi-harmonic free energy and molecular dynamics simulations to identify super-ionic behavior of various phases as a function of pressure and temperature. Be that as it may, the system that they considered is relevant to planetary science and this contribution will make an impact on the field. However, I would point out that some models of the planetary structure reproducing the gravitational moments of the Ice Giants already considered adding a helium (and hydrogen) component in the outer core of the planets [Figure 9 in N. Nettelmann, K. Wang, J. Fortney, S. Hamel, S. Yellamilli, M. Bethkenhagen, and R. Redmer, *Icarus* 275, 107 (2016)] since a lower density than what could be provided by the equation of state of water ammonia and methane mixtures was required to fit the gravitational moments of Uranus. While the authors point to that reference, their study lends support to that assumption and they should discuss this aspect in more detail. The computational details to reproduce the work are sufficient and clearly laid out.

Sebastien Hamel

Reviewer #2 (Remarks to the Author):

This manuscript describes the results of employing currently the Calypso code, the VASP code, and the Phonopy code to predict the structures and selected properties of possible helium-ammonia (He-NH₃) compounds.

The authors suggest that the predicted compounds are candidates for materials in the planets Neptune and Uranus. These computer codes employed are currently considered to be among the state-of-the-art codes for this type of study and the use of the computer codes as employed by the authors is considered to be standard practice for structure searches and characterization. All parameters employed for each code are adequately described and correctly chosen in this reviewers' opinion.

The molecular dynamics simulations that were also carried out to explore the superionic property of $(\text{NH}_3)_2\text{He}$ compound at high pressures and high temperatures are clearly described.

The characterization of He compounds is also correctly aided with the use of zero-point vibrational energy (ZPE). The Bader analysis discussion adds interesting detail to the characterization of the predicted compounds.

My specific comments for the authors to address and make appropriate minor modifications are listed below.

- 1) There is a sentence on page 4 that needs correction. It now is written as “: ”only H atoms in $(\text{NH}_4)^+$ unit diffuse around N atoms limited in a local space.” This needs to be clarified.
- 2) The total time for the ab initio MD should be added along with the time step already listed.
- 3) The reference list should be brought up-to-date. For example the reference “Peihao Huang, Hanyu Liu, Jian Lv, Quan Li, Chunhong Long, Yanchao Wang, Changfeng Chen, Russell J. Hemley, and Yanming Ma, PNAS March 17, 2020 117 (11) 5638-5643; H_3O in Neptune and Uranus” should be referenced and commented on.
- 4) The reference number 37 in the manuscript is incorrect and incomplete.
- 5) Can the authors make any predictive comment on the relative stability of compounds such as those references in references 36, 37, and that of the 2020 references mentioned above? The answer to this question may rely on future studies as suggested by the authors but any comments that the authors could make in the manuscript would be welcome
- 6) The manuscript contains too many minor English and grammatical deficiencies to be listed here. Another careful reading will be needed to eliminate these deficiencies.

In summary, although no new methods are developed, the methods employed are state-of-the-art so the results can definitely be considered to be reliable. To fully understand the composition of Neptune and Uranus, the authors have correctly employed current theoretical methods. Studies such as the one described in this manuscript provide new pieces of essential data to eventually characterize the composition of these planets. This manuscript may therefore provide a basic reference for expected future characterization studies of Neptune and Uranus.

Reviewers' comments:

Response to reviewer #1

REFeree#1 OVERVIEW. *The major claim of this paper is that there exists solid solutions of ammonia and helium at the pressure and temperature relevant to Ice Giants such as Neptune and Uranus and that it can support a super-ionic phase. The study is well constructed although somewhat derivative in that they basically follow the outline given in Bethkenhagen et al. (2015), i.e. a combination of structure search, quasi-harmonic free energy and molecular dynamics simulations to identify super-ionic behavior of various phases as a function of pressure and temperature. Be that as it may, the system that they considered is relevant to planetary science and this contribution will make an impact on the field. However, I would point out that some models of the planetary structure reproducing the gravitational moments of the Ice Giants already considered adding a helium (and hydrogen) component in the outer core of the planets [Figure 9 in N. Nettelmann, K. Wang, J. Fortney, S. Hamel, S. Yellamilli, M. Bethkenhagen, and R. Redmer, *Icarus* 275, 107 (2016)] since a lower density than what could be provided by the equation of state of water ammonia and methane mixtures was required to fit the gravitational moments of Uranus. While the authors point to that reference, their study lends support to that assumption and they should discuss this aspect in more detail. The computational details to reproduce the work are sufficient and clearly laid out.*

REPLY: We are grateful to the reviewer's positive comments on our manuscript. As mentioned by the reviewer, the model suggested in *Icarus* 275, 107 (2016) is really important, which firstly introduced helium and hydrogen into the outer core of the planets. In fact, this model together with the recent several works on pressure-stabilized helium-containing compounds motivate us the current study, and we are happy that our predicted stable He-NH₃ compounds at high pressure support

this model. We have added the introduction of this model and according discussions in the last paragraph on page 1 and the first paragraph of page 5, respectively.

Response to reviewer #2

REFeree#1 OVERVIEW. *This manuscript describes the results of employing currently the Calypso code, the VASP code, and the Phonopy code to predict the structures and selected properties of possible helium-ammonia (He-NH₃) compounds. The authors suggest that the predicted compounds are candidates for materials in the planets Neptune and Uranus. These computer codes employed are currently considered to be among the state-of-the-art codes for this type of study and the use of the computer codes as employed by the authors is considered to be standard practice for structure searches and characterization. All parameters employed for each code are adequately described and correctly chosen in this reviewers' opinion. The molecular dynamics simulations that were also carried out to explore the superionic property of (NH₃)₂He compound at high pressures and high temperatures are clearly described. The characterization of He compounds is also correctly aided with the use of zero-point vibrational energy (ZPE). The Bader analysis discussion adds interesting detail to the characterization of the predicted compounds. In summary, although no new methods are developed, the methods employed are state-of-the-art so the results can definitely be considered to be reliable. To fully understand the composition of Neptune and Uranus, the authors have correctly employed current theoretical methods. Studies such as the one described in this manuscript provide new pieces of essential data to eventually characterize the composition of these planets. This manuscript may therefore provide a basic reference for expected future characterization studies of Neptune and Uranus.*

REPLY: We are grateful for the referee's effort in a careful reading of the manuscript and the constructive comments. Following the referee's suggestions, we have made the proper revisions accordingly.

REFeree#2 COMMENT 1. *There is a sentence on page 4 that needs correction. It now is written as “: only H atoms in (NH₄)⁺ unit diffuse around N atoms limited in a local space.” This needs to be clarified.*

REPLY: In the structure of (NH₃)₂He, there exist three kinds of molecules NH₃, (NH₂)⁻ and (NH₄)⁺. At 100 GPa and 1000 K, molecular dynamic simulations show that H atoms in (NH₂)⁻ and (NH₄)⁺ keep vibrating around their lattice positions, while H atoms in NH₃ become diffuse. In the previous manuscript, we wrote carelessly “NH₃” to be “(NH₄)⁺”. The sentence has been rephrased to be “H atoms in NH₃ become diffuse while H atoms in (NH₂)⁻ and (NH₄)⁺ keep vibrating around their lattice positions” in the 1st paragraph of page 4.

REFeree#2 COMMENT 2. *The total time for the ab initio MD should be added along with the time step already listed.*

REPLY: Thanks for the referee’s suggestions. The *ab initio* MD simulations were performed consisting of 10,000 time steps with a time step of 0.5 fs. This has been added in the subsection “Molecular dynamics” on page 5.

REFeree#2 COMMENT 3. *The reference list should be brought up-to-date. For example the reference “Peihao Huang, Hanyu Liu, Jian Lv, Quan Li, Chunhong Long, Yanchao Wang, Changfeng Chen, Russell J. Hemley, and Yanming Ma, PNAS March 17, 2020 117 (11) 5638-5643; H₂O in Neptune and Uranus” should be referenced and commented on.*

REPLY: Thanks for reminding. The introduction of this work has been added in the 2nd paragraph of page 1 and the reference has been cited as Ref [28] in the revised manuscript.

REFeree#2 COMMENT 4. *The reference number 37 in the manuscript is incorrect and incomplete.*

REPLY: The previous Ref [37] has been corrected, which is cited as Ref [39] in the revised manuscript.

REFEREE#2 COMMENT 5. *Can the authors make any predictive comment on the relative stability of compounds such as those references in references 36, 37, and that of the 2020 references mentioned above? The answer to this question may rely on future studies as suggested by the authors but any comments that the authors could make in the manuscript would be welcome*

REPLY: We thank the referee for the suggestions. We have added such a comment in the 2nd paragraph of page 5 written as: “Previous studies have assumed the presence of NH₃, CH₄, H₂O, and H₂ inside the giant ice planets, our predicted stability of superionic (NH₃)₂He as well as the recent reported superionic H₂OHe₂ under the P-T conditions corresponding to the ice giants' upper mantles indicate that helium could be remained inside the planets during their formation. Coincidentally, the stability of NH₃-He and H₂O-He compounds provide an evidence to support the new three-layer model suggested by Nettelmann et al., in which helium was considered as a small component in outer core of the planets. Therefore, the current results are essential to the understanding of the interior models of these planets. Moreover, CH₄ and H₂ are another two main components in upper mantle of these planets, therefore, there is a highly possibility that helium could react with CH₄ or H₂ at high pressures to form new compounds, which deserves further investigation.”

REFEREE#2 COMMENT 6. *The manuscript contains too many minor English and grammatical deficiencies to be listed here. Another careful reading will be needed to eliminate these deficiencies.*

REPLY: We are grateful for the close reading of the manuscript by the referee and apologies that these typographic errors slipped through our previous version. We have corrected all spelling errors and tidied the text up in a number of places. We have also read the revised manuscript carefully to eliminate typographical and grammatical errors.